# Cap analysis of gene expression reveals alternative promoter usage in a rat model of hypertension

Sonal Dahale[1,2], Jorge Ruiz-Orera[3], Jan Silhavy[4], Norbert Hübner[3,5,6], Sebastiaan van Heesch[7] , Michal Pravenec[4], Santosh S Atanur[1,8]

The role of alternative promoter usage in tissue-specific gene expression has been well established; however, its role in complex diseases is poorly understood. We performed cap analysis of gene expression (CAGE) sequencing from the left ventricle of a rat model of hypertension, the spontaneously hypertensive rat (SHR), and a normotensive strain, Brown Norway to understand the role of alternative promoter usage in complex disease. We identified 26,560 CAGE-defined transcription start sites in the rat left ventricle, including 1,970 novel cardiac transcription start sites. We identified 28 genes with alternative promoter usage between SHR and Brown Norway, which could lead to protein isoforms differing at the amino terminus between two strains and 475 promoter switching events altering the length of the 5′ UTR. We found that the shift in *Insr* promoter usage was significantly associated with insulin levels and blood pressure within a panel of HXB/BXH recombinant inbred rat strains, suggesting that hyperinsulinemia due to insulin resistance might lead to hypertension in SHR. Our study provides a preliminary evidence of alternative promoter usage in complex diseases.

## Introduction

Mammalian gene transcription is tightly regulated. The core promoter, defined as a region ~40 bp upstream and downstream of the transcription start site (TSS) is sufficient for initiation of the transcription (Nepal et al, 2020). The general transcription initiation factors assemble at the core promoter with RNA polymerase II to form a multi-protein complex that facilitates accurate transcription (Nikolov & Burley, 1997). In mammalian genomes, although the number of protein-coding genes is limited, the transcript repertoire is much more diverse (Strausberg & Levy, 2007). It has been

estimated that more than 60% of the mammalian genes have multiple transcripts (Forrest et al, 2014). In part, the transcriptional diversity is achieved through the use of alternate promoters and alternate splicing (Strausberg & Levy, 2007). Most of the mammalian protein-coding genes are regulated by multiple promoters that initiate transcription for multiple gene isoforms (Carninci et al, 2006; Forrest et al, 2014). Alternate splicing regulates gene isoform expression post transcriptionally. On the contrary, alternate promoters provide a way to regulate gene isoform expression pre transcriptionally (Ayoubi & Van De Yen, 1996; Demircioğlu et al, 2019).

The transcriptional diversity due to alternate promoters could be assessed by precisely mapping 5′ ends of the transcripts. Cap Analysis of Gene Expression (CAGE), which takes advantage of the 7-methylguanosine cap structure of the transcripts, allows accurate genome-wide mapping of TSSs at single base-pair resolution (Carninci et al, 2005, 2006). Using CAGE, the FANTOM consortium has mapped precise TSSs across multiple tissues and primary cells from mouse and humans (Forrest et al, 2014). Approximately, 80% of the CAGE-defined TSS showed tissue-specific expression, suggesting that most mammalian genes use alternative promoters in a tissue-specific manner to regulate tissue-specific gene expression (Forrest et al, 2014). Furthermore, it has been shown that many transcripts switch promoters between oocytes and the zygotic stage during embryonic development in zebra fish (Haberle et al, 2014). Often this shift in TSS usage is within 100 bp of the same promoter. Although alternative promoter usage between different tissues and cell types, and between various developmental stages is well established (Forrest et al, 2014; Haberle et al, 2014), the role of alternate promoter usage in disease has not been studied extensively.

The spontaneously hypertensive rat (SHR) strain is one of the most widely used animal models to study hypertension, which also shows many metabolic phenotypes, including insulin resistance, dyslipidemia, and central obesity, collectively known as metabolic syndrome (MtS) (Atanur et al, 2010). By crossing SHR with the

[1]Department of Metabolism, Digestion, and Reproduction, Faculty of Medicine, Imperial College London, London, UK  [2]Department of Microbial Sciences, Faculty of Health and Medical Sciences, University of Surrey, Guildford, UK  [3]Cardiovascular and Metabolic Sciences, Max Delbrück Center for Molecular Medicine in the Helmholtz Association (MDC), Berlin, Germany  [4]Institute of Physiology of the Czech Academy of Sciences, Prague, Czech Republic  [5]Charité -Universitätsmedizin, Berlin, Germany  [6]DZHK (German Centre for Cardiovascular Research), Partner Site Berlin, Berlin, Germany  [7]Princess Máxima Center for Pediatric Oncology, Utrecht, Netherlands  [8]The National Institute for Health Research, Imperial Biomedical Research Centre, ITMAT Data Science Group, Imperial College London, London, UK

Correspondence: santosh.atanur@imperial.ac.uk

normotensive Brown Norway (BN) strain, the HXB/BXH panel of recombinant inbreed (RI) rat strains have been derived (Pravenec et al, 2014). Large numbers of physiological quantitative trait loci (pQTLs), expression QTLs (eQTLs), and histone QTLs (hQTLs) have been mapped using RI strains (Hubner et al, 2005; Petretto et al, 2006; Aitman et al, 2008; Johnson et al, 2014; Rintisch et al, 2014). Whole genome sequencing of both the parental strains SHR and BN have revealed more than four million genomic variants between the two strains (Atanur et al, 2010, 2013; Simonis et al, 2012). Availability of a rich resource of phenotypic and genotypic data and homozygosity throughout the genome provides a unique opportunity to perform genetic studies in SHR.

We performed CAGE tag sequencing from left ventricles (LVs) of SHR and BN rat strains. Along with precisely mapping TSSs in rat heart, we also show that a large number of genes use alternate promoters between the SHR and BN strains, suggesting a role of alternative promoter usage in complex disease.

# Results

### Identification of CAGE-defined promoters in rat heart

To understand the role of alternate promoter usage and promoter shift in complex diseases, we performed CAGE tag sequencing from the LVs of the rat model of hypertension SHR and a normotensive rat strain BN in three replicates each from both male and female rats. We performed "non-amplifying, non-tagging Illumina CAGE" (nAnT-iCAGE) because no tagging protocol allows the sequencing of longer reads (100 bp) and a non-amplification protocol eliminates biases due to PCR amplification (Murata et al, 2014). On average we sequenced 15 million uniquely mapped read pairs per sample. To avoid mapping bias due to genomic variants between SHR and BN rat strains, we created a pseudo-SHR genome by substituting all the SHR single nucleotide variants in the BN reference genome (Gibbs et al, 2004). CAGE tags from SHR were mapped to the pseudo-SHR genome, whereas BN CAGE tags were mapped to the BN reference genome. To obtain a comprehensive list of the rat heart promoters, we performed an analysis of all 12 samples together. We identified a total of 26,560 tag clusters, each representing a unique promoter region from the rat heart. To assess the reproducibility of the experiments we estimated all possible pairwise correlations between the three biological replicates of each sample (SHR male, SHR female, BN male, and BN female). The biological replicates of each sample showed a pairwise Pearson correlation coefficient of 0.98 or higher (Fig S1A–D), suggesting that our results are highly reproducible.

CAGE-defined promoters were annotated by overlapping them with the rat gene annotations from ENSEMBL (Version 93) (Howe et al, 2021). We defined a gene region as a region between the start and end positions of the longest transcript of a protein-coding gene plus 1 kb upstream region. Approximately, 80% (n = 21,353) of the CAGE-defined promoters were located in the gene regions of 10,935 protein coding genes (Fig 1A). We found that, in rat heart, a total of 6,445 genes use a single promoter, whereas 4,490 genes use more

than one promoter (Fig S2). For genes that use more than one promoter, the promoters were ranked based on their expression levels (TPM) as promoter 1 (P1), promoter 2 (P2), and so on.

### Comparison with the human, mouse, and rat CAGE tag data from FANTOM consortium

Using CAGE, the FANTOM consortium has mapped precise TSSs across multiple tissues and primary cells of mouse and humans (Forrest et al, 2014). In addition, the FANTOM consortium has performed CAGE tag sequencing of three cell types (Aortic smooth muscle cells, hepatocytes, and mesenchymal stem cells) and a tissue from rat (Lizio et al, 2017). We performed a systematic comparison between the rat heart CAGE tag clusters identified in this study with the CAGE data from the FANTOM consortium. A total of 41.13% (n = 10,924) rat heart CAGE tag clusters overlapped with the FANTOM rat CAGE tag clusters. The FANTOM consortium rat CAGE data have been generated from four non-cardiac cell types/tissues from Sprague–Dawley and Lewis rat strains. The small overlap of rat heart CAGE data (this study) and the FANTOM rat CAGE data could be due to the differences in the strains and the cell types/tissues used in both studies. This result, therefore, highlights the necessity of performing CAGE tag sequencing of a large number of tissues and cell types to obtain a comprehensive list of transcriptome repertoire of the rat, one of the widely used rodent models of disease.

To compare the rat heart CAGE data with the human and mouse CAGE data from the FANTOM consortium, we first identified human (hg19) and mouse (mm9) genomic regions orthologous to the rat heart CAGE-defined promoter regions (see Methods). A total of 15,729 (59.22%) rat heart CAGE-defined promoter regions overlapped with the mouse heart CAGE tag clusters, whereas 12,387 (46.64%) rat heart promoters overlapped with the human heart promoters identified by the FANTOM consortium using CAGE. In addition, a small proportion (n = 140 and n = 214) of the rat heart CAGE tag clusters overlapped with the mouse and human CAGE promoters from the non-cardiac tissues. This result suggests the significant conservation of promoter usage between humans, mouse, and rat heart.

### Characterisation of rat heart CAGE tag clusters

The promoters can be broadly classified into two categories: TATA box–rich and GpG-rich promoters. More than half of the mammalian genes initiate transcription from GpG dinucleotide-rich region referred to as CpG islands (Vavouri & Lehner, 2012). Moreover, the housekeeping genes are very often preceded by the CpG islands. It has been shown that the TATA-rich promoters tend to be sharper with most of the CAGE signal coming from a few nucleotides at the promoter, whereas CpG-rich promoters tend to be much broader with CAGE signal distributed across relatively larger promoter region (Carninci et al, 2006). Hence, we investigated rat heart CAGE-defined promoters for the presence of CpG islands and TATA box. A total of 13,005 (48.96%) rat heart promoters overlapped with the CpG islands, whereas 8,695 (32.73%) promoters had TATA box. We further classified rat heart promoters into three classes; promoters that contain only TATA box (n = 5,190, 19.94%), only CpG island (n = 9,500, 35.77%), and the promoters that contain both CpG

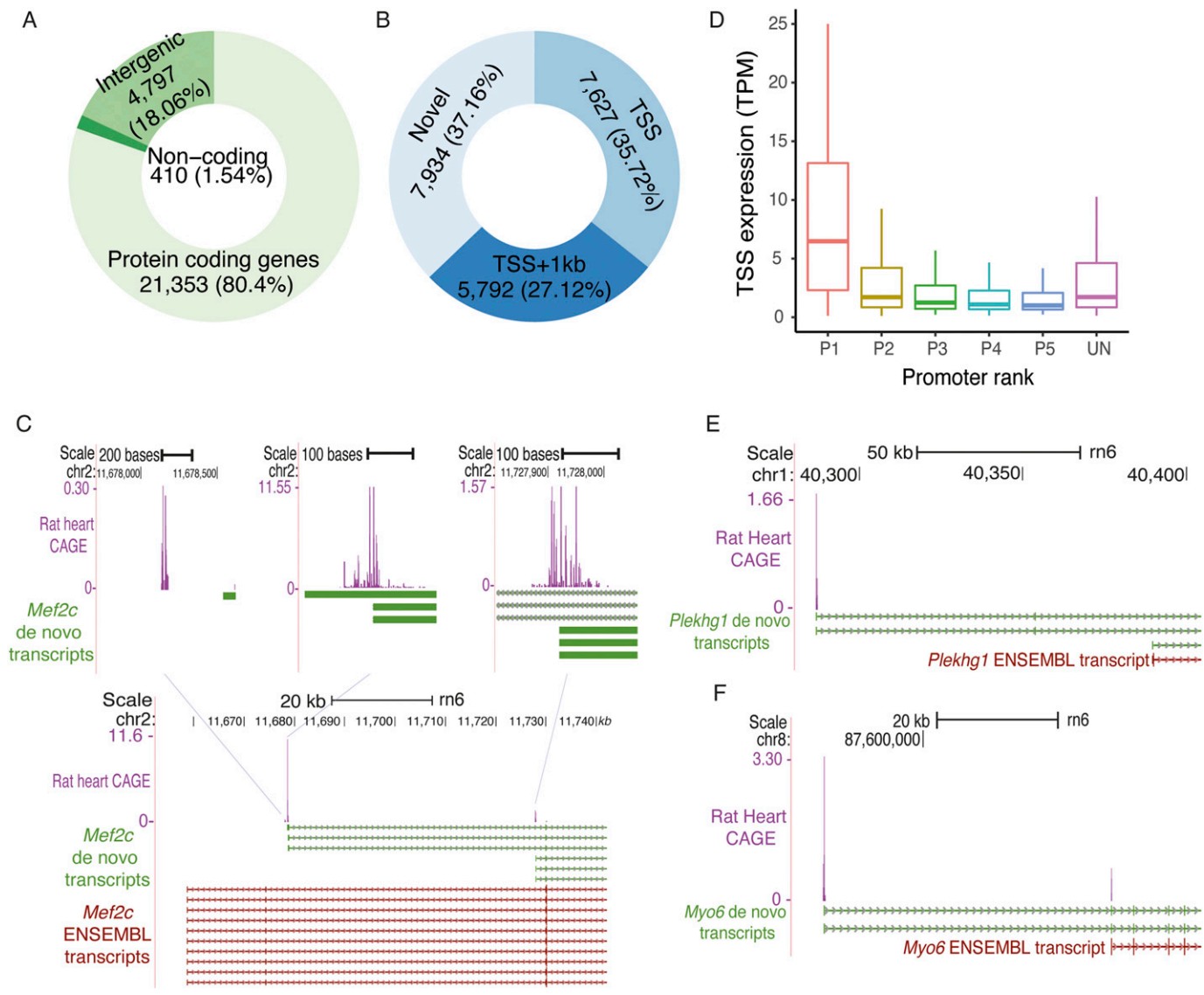

**Figure 1. CAGE tag sequencing identifies promoters from rat left ventricle.**
**(A)** Annotations of the CAGE-defined promoters to genomic features. **(B)** Classification of CAGE-defined transcription start site located in the gene regions, defined as the region between transcription start and end of the longest transcript of the gene plus 1 kb upstream region, on the same strand as that of the gene. **(C)** Example of novel heart-specific transcripts for identified using CAGE tag sequencing for gene *Mef2c*. **(D)** Distribution of expression levels at promoters by promoter rank including the promoters that are unassigned to any protein-coding gene. **(E, F)** Examples of novel transcription start site annotations for gene *Plekhg1* and *Myo6*, respectively. Source data are available for this figure.

island and TATA box (n = 3,505, 13.20%). The TATA box only promoters were significantly shorter in size than the CpG-rich promoters as well as promoters that contain both CpG and TATA box (Mann–Whitney test $P < 2.2 \times 10^{-16}$ and $P < 2.2 \times 10^{-16}$, respectively; Fig S3A and B). However, there was no difference in length of CpG only promoters and the promoters with both CpG islands and TATA box (Mann–Whitney test $P = 0.4066$, Fig S3A and B).

## CAGE tag sequencing improves rat TSS/transcripts annotations

We evaluated CAGE-defined TSSs against the ENSEMBL TSS annotations for the rat genome. Of 21,353 CAGE-defined TSS that were

within the gene regions, only 35.71% (n = 7,627) matched perfectly with the ENSEMBL TSS annotations. Additional 1,965 tag clusters were within the 100 bp of ENSEMBL annotated TSS, whereas 3,827 tag clusters were between 100 bp to 1,000 bp away from the ENSEMBL annotated TSS. Surprisingly, no TSS annotations were found in ENSEMBL for the remaining 37% (n = 7,934) tag clusters that were located within the protein coding gene regions and were on the same strand as that of the gene (Fig 1B). Most of these CAGE tag clusters might represent promoters of the cardiac transcripts that are not annotated in ENSEMBL. To investigate this hypothesis, we performed a de novo assembly of publicly available RNA-seq data from LVs of SHR and BN rats (Johnson et al, 2014). Of 7,934 tag

clusters with no ENSEMBL TSS annotation, 383 overlapped perfectly with the TSS of de novo transcripts, whereas 857 tag clusters were located within 500 bp of de novo transcript TSSs (Table S1), suggesting that some of these tag clusters indeed represent promoters of rat heart transcripts.

Next, we investigated whether genic rat heart CAGE tag clusters without ENSEMBL TSS annotations were represented in other rat cell types/tissues or in cardiac tissue from human and mouse. Of 7,934 tag clusters, a small proportion of tag clusters (n = 730) were also present in FANTOM rat CAGE data. Furthermore, 2,326 rat heart promoters overlapped with mouse heart promoters, and 2,177 overlapped with human heart promoters. In total 3,317 (41.81%) rat heart tag clusters located within the gene regions with no ENSEMBL TSS annotations overlapped with CAGE-defined promoters either from other rat tissues, human heart, or mouse heart. This suggests that most unannotated genic rat heart CAGE tag clusters indeed represent ubiquitous and/or conserved cardiac TSSs.

A gene myocyte enhancer factor 2 (*Mef2c*), a member of the MEF2 family of transcription factors, is a key regulator of cardiovascular development (Materna et al, 2019). Loss-of-function (LoF) mutations in *MEF2C* have been implicated in dilated cardiomyopathy in humans (Yuan et al, 2018). ENSEMBL annotations for *Mef2c* in the rat genome show 10 different transcripts but all of them contain the same TSS (chr2:11,658,568). Interestingly, no CAGE tag cluster from rat heart overlap with the ENSEMBL annotated TSS of *Mef2c*. On the contrary, three CAGE-defined tag clusters were located within the *Mef2c* gene body. The de novo transcripts assembly identified six transcripts for *Mef2c* with three distinct TSSs, all of them overlapped with the CAGE-defined TSSs (Fig 1C). The TSS of the *Mef2c* transcript that shows the highest expression (58.44 TPM) at the promoter was located 20 kb downstream to the ENSEMBL annotated TSS. The *Mef2c*_P1 was present in FANTOM rat cage data, whereas all three rat *Mef2c* promoters were observed in mouse and two of them were present in human CAGE data. Human and mouse orthologs were annotated as *Mef2c* promoters in both species. Furthermore, their activity levels were also conserved as mouse and human orthologous regions of predominant (P1) rat *Mef2c* promoter were also the most predominant promoter in both species. However, in ENSEMBL, none of the three CAGE tag clusters were annotated as a *Mef2c* TSS. This suggests that our CAGE data could help in improving TSS annotations of important cardiac genes in rat.

A total of 5,207 tag clusters were located in intergenic regions of the genome, of which 410 were associated with annotated non-coding RNAs. We compared the expression levels of the remaining 4,797 intergenic tag clusters with the expression levels of coding gene promoters ranked based on the expression. The average expression levels of intergenic tag clusters ranged between the average expression of the strongest promoter of genes (P1) and the second strongest promoter of the gene (P2; Fig 1D), suggesting that some of the intergenic tag clusters might also be promoters of the protein-coding genes. To further investigate intergenic tag clusters, we compared them with the de novo transcripts identified using RNA-seq data from rat heart. A total of 827 intergenic tag clusters were located within 100 bp distance of de novo transcript TSSs. In addition, 286 were within 500 bp of de novo transcripts (Table S2). Furthermore, 3,054 (63.66%) rat heart intergenic CAGE tag clusters overlapped with the CAGE-defined promoters either from other rat

tissues (n = 1,229), mouse heart promoters (n = 2,578), or human heart promoters (n = 1,832) when compared with FANTOM CAGE dataset. This suggests that a large proportion of rat heart intergenic tag clusters are indeed a TSSs of a rat transcripts; however, they were not annotated as a TSS in ENSEMBL.

A total of 1,743 intergenic rat heart CAGE tag clusters were novel for which no TSS annotation was found in ENSEMBL rat gene annotations or any of the FANTOM CAGE datasets (rat, mouse, and human). For gene *Plekhg1*, no CAGE tag cluster was found at the ENSEMBL annotated TSS. The nearest CAGE tag cluster to gene *Plekhg1* was more than 100 kb upstream to ENSEMBL TSS, which was supported by our de novo transcript assembly (Fig 1E). For the *Myo6* gene, only one transcript was annotated in ENSEMBL, TSS of which was supported by the CAGE tag cluster. However, CAGE data and de novo transcript assembly identified another transcript for the *Myo6* gene with a TSS around 47 kb upstream of the ENSEMBL annotated TSS (Fig 1F). This novel transcript showed significantly higher expression (20.57 TPM) as compared to the ENSEMBL annotated transcript (0.96 TPM). Furthermore, the annotated transcript was up-regulated in SHR as compared with BN, whereas the novel transcript was down-regulated in SHR as compared with BN, suggesting alternative use of two *Myo6* transcripts in SHR and BN.

In summary, our results show that rat heart transcript annotations could be significantly improved using CAGE data by precisely mapping TSSs.

### CAGE identifies alternative promoter usage between SHR and BN

To understand the role of alternative promoter usage in disease, we explored CAGE tag data from SHR and BN rat strains. We hypothesised that if a gene is predominantly transcribed from two different promoters in SHR and BN, respectively, then the tag clusters associated with both promoters should show differential expression but with expression difference in the opposite direction. A total of 4,490 (41%) of the heart-expressed genes showed more than one CAGE-defined promoter. We selected the promoters that showed a minimum of 20% expression of the total expression of all promoters of a gene. The majority (3,571) of the genes had only one predominant promoter, whereas 918 genes had at least two promoters with more than 20% activity. Of these 918 genes, for 471 genes none of the promoters show any difference in expression between SHR and BN, whereas for 447 genes at least one promoter showed statistically significant differential expression. For 419 genes, all the promoters associated with a gene showed expression differences in the same direction, whereas for 28 genes, at least two promoters showed differential expression in the opposite direction (Fig 2A), suggesting that these 28 genes use alternative promoters between SHR and BN. For all these genes, the use of an alternative promoter between SHR and BN leads to a shorter transcript in one of the two strains with an impact on the protein coding region of the gene. Of 28 genes that use alternative promoter between SHR and BN in the heart, both the alternative promoters of 10 genes were conserved in humans, whereas only one of the two promoters of eight genes was conserved in humans. Similarly, both the alternative promoters of 13 genes were conserved in mouse, whereas for nine genes, only one promoter was conserved in mouse.

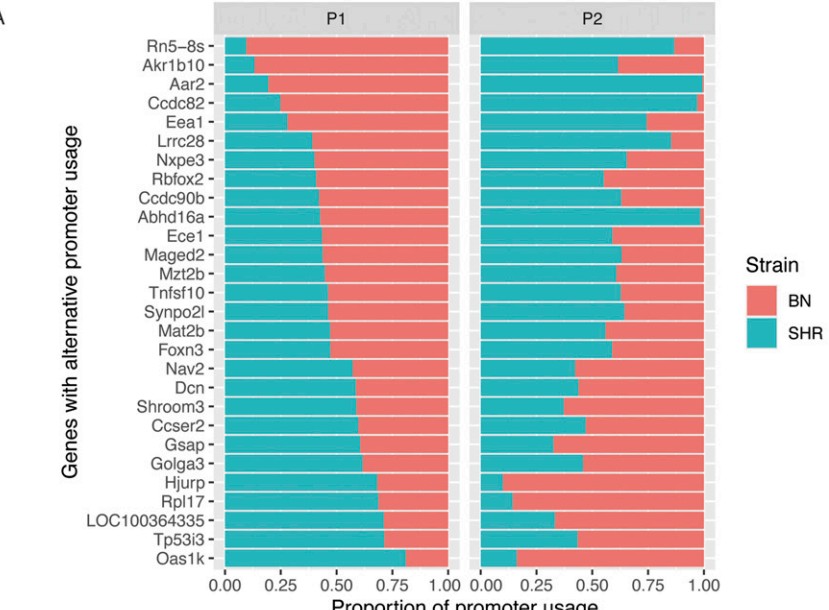

**Figure 2. Alternative promoter usage between spontaneously hypertensive rat (SHR) and Brown Norway (BN).**
**(A)** List of genes with alternative promoter usage between SHR and BN. The P1 panel represents the first promoter, whereas the P2 panel represents the second promoter. The x-axis represents the proportion of the promoter used in SHR (green) and the BN (orange). **(B)** Example of alternative promoter usage between SHR and BN in *Shroom3* gene. In SHR *Shroom3* is predominantly transcribed from the first promoter, whereas in BN it is predominantly transcribed from the second promoter. The gene *Shroom3* is located on a negative-strand hence promoter expression levels have negative values. Source data are available for this figure.

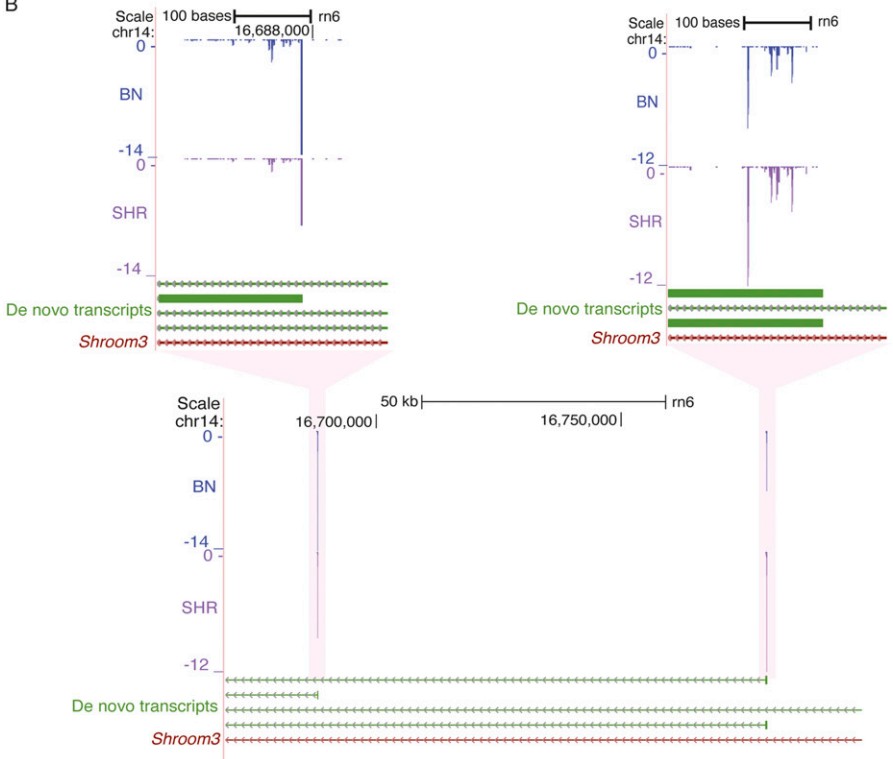

These 28 genes include *Synpo2l, Abhd16a, Ece1, Shroom3* (Fig 2B), and *Rbfox2* all of which are implicated in cardiovascular disorders including hypertension in humans (Yasuda et al, 2007; Nutter et al, 2016; Xu et al, 2018; Durbin et al, 2020; Clausen et al, 2021). Loss of function (LoF) mutations in human *SHROOM3 and SYNOP2L* genes have been shown to be associated with congenital heart defects (Durbin et al, 2020) and arterial fibrillation

(Clausen et al, 2021), respectively. Dysregulation of *RBFOX2* has been shown to be an early event in cardiac pathogenesis of diabetes in human (Nutter et al, 2016). The polymorphisms in the *ECE1* gene have been implicated in human essential hypertension (Jin et al, 2003; Yasuda et al, 2007), whereas *ABHD16A* is shown to be associated with coronary artery aneurism (Hsieh et al, 2010; Xu et al, 2018).

## Promoter shift between SHR and BN rat strains

It has been shown that the switch in TSS usage within the same promoter, often within 100 bp, happens between maternal and the zygotic transcript during zebra fish development (Haberle et al, 2014). We investigated whether TSS switching happens in a disease condition by using SHR and BN CAGE tag data. We identified 475 transcripts with a shift in TSS usage within the same promoter between SHR and BN, of which 287 (60%) were dominant promoters (P1) of the genes. However, only 50% of them were assigned to the ENSEMBL annotated genes, whereas the remaining 50% were novel transcripts. In most cases, the TSS shift happened within 100 bp of the same promoter, as observed previously (Haberle et al, 2014), and the shift happened in both directions with respect to reference rat strain (BN; Fig 3A). The ENSEMBL annotated genes that show a shift in TSS usage between SHR and BN were enriched for genes involved in metabolic processes ($P = 0.003$) and chloride transport ($P = 0.005$). The genes that showed a switch in TSS usage within the same promoter between SHR and BN include *Insr*, *Endog*, *Vnn1*, and *Serpina3c*.

Of 475 heart transcripts that showed a shift in the TSS usage between SHR and BN, human orthologous regions of 182 promoters showed CAGE tag signals in FANTOM data. Of these 182 transcripts, for 89 transcripts only one of the two (SHR or BN) promoters was represented in humans, whereas both SHR and BN promoters were represented in humans for 93 transcripts. Similarly, a total of 200 rat shifting promoters were conserved in mouse. In mouse, 144 transcripts showed both SHR and BN promoters, whereas 56 mouse transcripts showed only one of the two (SHR or BN) promoters.

The transcripts in which the shift in TSS usage between two rat strains was observed produce a longer transcript in one strain compared with a shorter transcript in another, in most cases affecting the length of the 5′ UTR. Being inbred rat strains, SHR and BN are completely homozygous, hence the F1 cross between SHR and BN rat strains tend to be heterozygous at all the variable loci between SHR and BN. The F1s inherit both longer and shorter transcripts of the genes that show promoter switching between the two rat strains, thus both the promoters show intermediate expression in F1s (Fig S4A–C). We reasoned that the genomic variants located in regions that are unique to longer transcripts should show a strong allelic imbalance in F1s. To test this hypothesis, we generated reciprocal F1 crosses by crossing SHR females with BN male and BN female with SHR male. For each reciprocal cross, we performed CAGE tag sequencing of LVs from three male and three female rats, totaling 12 F1 rats.

A total of 1,853 genomic variants between SHR and BN were located in CAGE tag clusters (CAGE-defined promoters), of which 110 variants were located in shifting promoters, remaining in normal promoters. The variants located in shifting promoters showed a strong allelic imbalance in F1s (Fig 3B) as compared with variants located in the non-shifting, non-differentially expressed promoters (Fisher's exact test, $P < 2.2 \times 10^{-16}$). In F1s, allelic imbalance in shifting promoters was comparable to the allelic imbalance of variants located in promoters that were strongly differentially expressed (adjusted $P$-value $\leq 0.05$ and $\log_2$ fold change > 1) in parental strains (SHR and BN; Fig 3C). As expected, the variants located in non-differentially expressed promoters did not show allelic imbalance

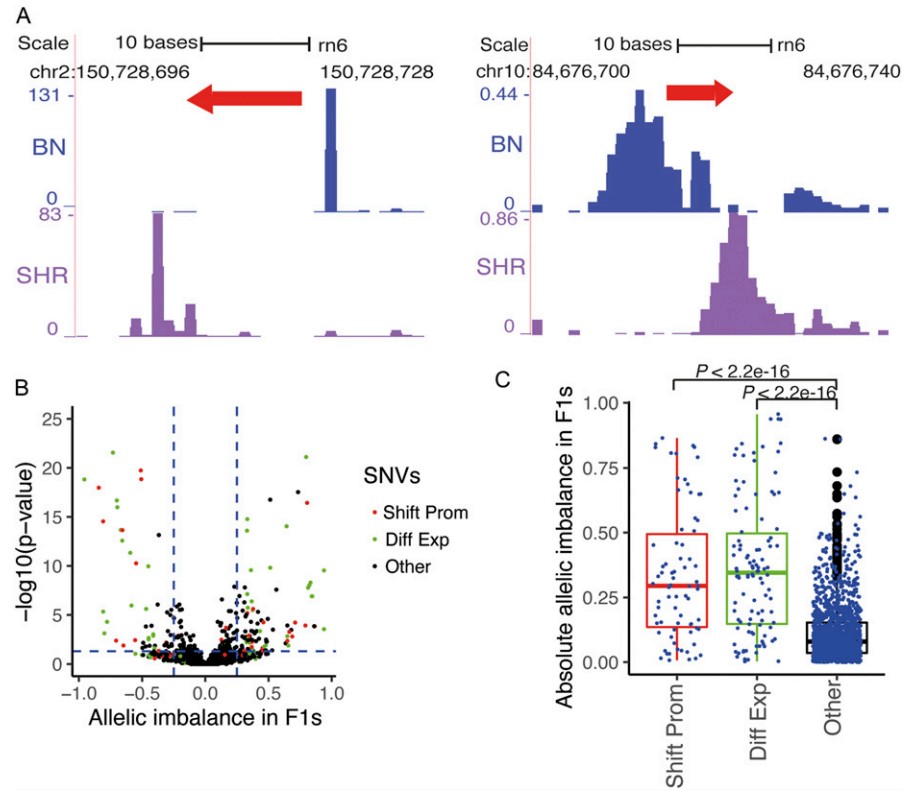

**Figure 3. Shifting promoters between spontaneously hypertensive rat (SHR) and Brown Norway (BN).**
**(A)** Shift in transcription start site usage within the same promoter happens in both directions with respect to reference BN strain. **(B)** Allelic imbalance in F1 crosses obtained by crossing SHR and BN. The x-axis represents allelic imbalance. The center (zero) indicates no allelic imbalance between SHR and BN allele in F1. Positive values indicate allelic imbalance towards in reference (BN) allele, whereas negative values indicate allelic imbalance towards SHR allele. The horizontal blue dotted line corresponds to a *P*-value equal to 0.05, whereas the vertical blue dotted line corresponds to the allelic imbalance of 0.25 and −0.25. The red dots represent variants located in shifting promoters, whereas green dots represent variants in promoters that show differential expression between parental strains (SHR and BN). Black dots represent genomic variants in CAGE-defined promoters that are neither shifting promoters nor differentially expressed between parental strains. **(C)** Box plot showing the distribution of allelic imbalance in F1 crosses in shifting promoters, differentially expressed promoters, and remaining CAGE-defined promoter regions. Shifting promoters (red box) show significantly (Mann–Whitney test) higher proportion variants with allelic imbalance as compared to non-shifting and non-differentially expressed promoters (black box). Shift Prom: shifting promoters, Diff Exp: differentially expressed promoters in parental strains (SHR and BN), Other: CAGE-defined promoters that are neither shifting promoters nor differentially expressed in parental strains.
Source data are available for this figure.

(Fig 3B and C). The strong allelic imbalance of variants located in shifting promoters confirms the presence of shifting promoters between SHR and BN rat strains.

### Promoter shift in *Insr* gene between SHR and BN

The insulin signaling pathway plays a key role in metabolic regulation, growth control, and neuronal function (Belfiore et al, 2017). This pathway is mediated by the insulin receptor (*Insr*), a transmembrane protein with tyrosine kinase activity (Payankaulam et al, 2019). We found that *Insr* shows a shift in TSS usage between SHR and BN (Fig 4A). In the BN rat strain, transcription starts predominantly from genomic position chr12:1,816,432, which was 346 bp upstream to the *Insr* start codon (chr12:1,816,086). In SHR rat strain, the TSS was observed 414 bp upstream (chr12:1,816,500) to the *Insr* start codon, resulting in a longer transcript in SHR with extended 5′ UTR as compared with BN. To validate this observation, we investigated RNA-seq data from SHR and BN heart. No RNA-seq reads were observed in the BN heart in the extended region that was specific to SHR, supporting the findings from CAGE data (Fig 4A). The promoter of rat *Insr* gene had a signature of a housekeeping gene with a CpG island at the promoter region same as the human *INSR* promoter region (Araki et al, 1989; McKeon et al, 1990). Both BN and SHR CAGE-defined TSSs were located within the CpG island.

A genomic variant (g.1816552:A>G) was located in the SHR specific region of *Insr* transcript. This variant showed significant allelic imbalance (Binomial test, $P = 2.77 \times 10^{-15}$) in F1 crosses with most reads overlapping the SHR specific region showed the SHR allele at the variant position (Fig 4B), suggesting that the longer transcript was inherited from SHR in F1s. This finding confirms the shift in TSS usage between SHR and BN for the gene *Insr* in the heart. The HXB/BXH panel of recombinant inbred (RI) rat strains, generated by crossing SHR and BN rat strains, have been widely used for genetic mapping (Kuneš et al, 1994). Recently, RNA-seq has been performed

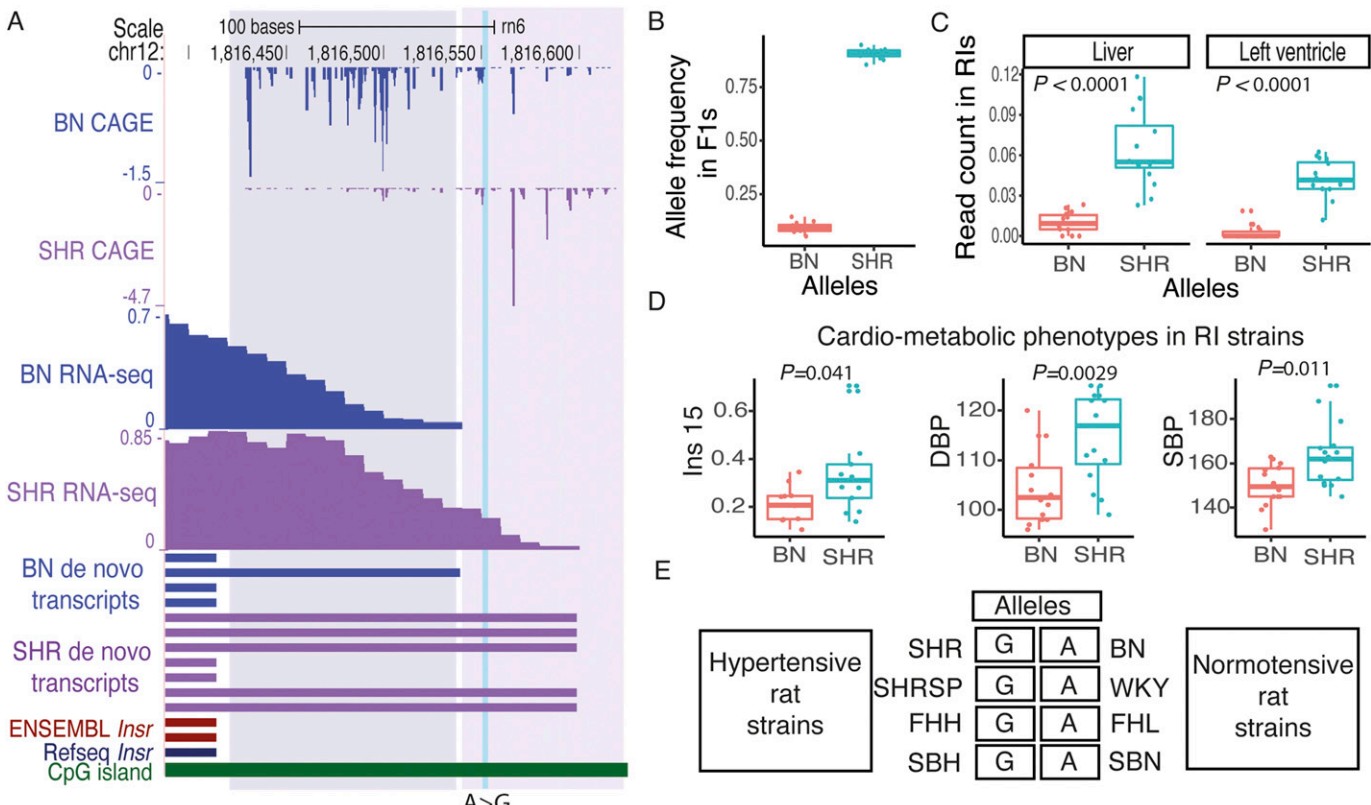

**Figure 4.   Shifting promoter in insulin receptor (*Insr*) gene.**
**(A)** Shifting promoter in *Insr*. The GAGE tag sequencing identified promoter shift between spontaneously hypertensive rat (SHR) and Brown Norway (BN) strain for *Insr* gene which leads to longer *Insr* transcript in SHR as compared with BN. Finding from CAGE was confirmed by RNA-seq data from rat left ventricle (LV) as well as de novo transcript assembly generated using rat LV RNA-seq data. **(B)** The variant g.1816552:A>G located in SHR specific *Insr* transcript region shows significant allelic imbalance in F1 crosses with most reads showing the SHR allele in F1s. **(C)** Based on the genotype at g.1816552:A>G recombinant inbred (RI) strains were grouped into two groups, where AA represents RI strains inheriting BN allele, whereas GG represents RI strains inheriting SHR allele. RNA-seq read count in SHR specific *Insr* transcript region was normalised to the read count from the first exon of *Insr* gene using publicly available RNA-seq data from left ventricle and liver in RI strains. Both in the LV and liver, RI strains that inherited SHR allele showed higher normalised reads count suggesting that they harbor longer transcript for *Insr* gene, whereas RI strains that inherited BN allele use the shorter transcript. **(D)** RI strains that harbor SHR allele showed significantly higher levels of insulin, systolic, and diastolic blood pressure. **(E)** Genotype in rat models of hypertension and their control strains at variant (g.1816552:A>G) located in SHR specific *Insr* transcript region. All hypertensive rat strain contains GG allele, same as SHR, whereas all the normotensive rat strains contain AA allele, same as BN. SHR, spontaneously hypertensive rat; BN, Brown Norway; ins15, insulin concentrations in 10-wk-old male rats fed a diet with 60% fructose from 8 wk to 10 wk (15 d), SBP, systolic blood pressure; DBP, diastolic blood pressure.
Source data are available for this figure.

in heart and liver tissues from RI strains (Witte et al, 2021). Using the variant g.1816552:A>G we grouped RI strains into two groups, depending on whether the SHR or the BN allele was inherited. The RI strains that inherited the SHR allele predominantly used a longer transcript both in the heart and liver (Fig 4C).

Next, we investigated several the cardio-metabolic phenotypes measured in RI strains (Pravenec et al, 2002; Kuneš et al, 2008). We found a significant association between the RI strain genotype at g.1816552:A>G and the insulin level (Mann–Whitney test, $P$ = 0.041), systolic blood pressure (Mann–Whitney test, $P$ = 0.0029), and diastolic blood pressure (Mann–Whitney test, $P$ = 0.011) in RI strains. The RI strains containing the SHR allele (GG; long transcript) showed significantly elevated levels of insulin, systolic and diastolic blood pressure (Fig 4D). Interestingly, a large number of physiological quantitative trait loci (pQTLs) for blood pressure and insulin resistance have been mapped to the region harboring *Insr* (Rat Genome Database; http://rgd.mcw.edu/).

Multiple rat strains have been developed to study hypertension and the whole genome of most of the rat models of hypertension have been sequenced (Atanur et al, 2013). We found that multiple rat models of hypertension; Fawn Hooded Hypertensive, Sabra Hypertensive, and Spontaneously Hypertensive Rat Stroke Prone, also harbor SHR allele at *Insr* promoter, whereas respective control strains Fawn Hooded Low blood pressure, Sabra Normotensive, and Wistar Kyoto contain BN allele (Fig 4E). Our results indicate that there is a strong association between longer *Insr* transcript, and the cardio-metabolic phenotype shown by these rat strains.

Taken together, multiple lines of genetic evidence suggest that the switch in TSS usage for the insulin receptor (*Insr*) gene in SHR might be associated with the disease phenotypes observed in the SHR rat strain. However, strong experimental support is required to establish a causal relationship between differential *Insr* TSS usage in SHR and the disease phenotype.

# Discussion

Cap analysis of gene expression (CAGE) has been widely used to precisely map TSSs in humans and mouse (Carninci et al, 2006; Forrest et al, 2014). Using CAGE it has been shown that the promoter usage tends to be tissue and developmental stage specific (Forrest et al, 2014; Haberle et al, 2014). Ubiquitously expressed genes achieve cell type specificity through the use of cell type–specific TSS (Feng et al, 2016). Although the role of alternate promoter usage has been well established in tissue-specific expression of genes, its role in disease has been poorly understood. To understand the role of alternate promoter usage in complex disease, we performed CAGE tag sequencing of LVs of two rat strains, SHR, a widely used model to study hypertension, and normotensive rat strain BN, progenitors of HXB/BXH RI strains.

Rat is a widely used animal model to study various disease phenotypes specifically cardio-metabolic phenotypes (Aitman et al, 2008). However, to date the precise TSSs of rat genes have been mapped only in three cell types (Aortic smooth muscle cells, hepatocytes, and mesenchymal stem cells) and a tissue (Lizio et al, 2017). In this study, we show that CAGE tag sequencing from rat heart (LVs) significantly improves the TSS annotations of the rat transcriptome. Using combined analysis of de novo transcript assembly and CAGE tag sequencing we show that TSSs of 1,113 transcripts were located more than 1,000 bp upstream to the annotated TSSs. In addition, we identified a large number of cardiac TSSs that were not annotated in ENSEMBL, this included three TSSs for myocyte enhancer factor 2 (*Mef2c*). In this study, we could identify thousands of novel TSS/transcripts even though we performed CAGE tag sequencing from only one rat tissue. This suggests that CAGE tag sequencing from a wide range of rat tissues could significantly improve rat TSS/transcript annotations. In addition, it will help to understand tissue-specific usage of TSS in rats and it would facilitate comparative analysis with human and mouse CAGE data.

Rat heart CAGE data have been generated from the whole LV. The heart is a complex organ with various cell types (Talman & Kivelä, 2018). In absence of single-cell transcriptomic data from rat hearts, it is challenging to determine cell type specificity of the CAGE tag signal. Cardiomyocytes and endothelial cells are the two most abundant cardiac cell types (Talman & Kivelä, 2018) and human heart single-cell transcriptomic data suggest that 50% of the ventricular cells tend to be cardiomyocytes (Litviňuková et al, 2020). We, therefore, speculate that most CAGE-defined TSSs might be active in cardiomyocytes. However, small proportions of TSS signals might also be coming from other rat left ventricular cell types such as endothelial cells, fibroblast, pericytes, and smooth muscle cells.

To understand the role of alternative promoter usage in disease, we used CAGE tag data from SHR and BN rat strains. We identified two types of TSS switching events between SHR and BN. First, where genes use completely different promoters between two rat strains that are located more than 100 bp away from each other and affect the coding region of the gene. This even leads to a shorter protein sequence because of truncated N terminus in one strain compared with the other. Both the transcripts show expression in both strains; however, one transcript tends to be predominantly expressed in one strain, whereas the other transcript shows predominant expression in the other strain. We identified 28 genes with alternative promoter usage between SHR and BN, a proportion of them were known to be associated with cardiovascular disorders. This suggests that alternative promoter usage could potentially result in a complex disease phenotype; however, extensive experimental validations are required to confirm these findings.

In the second type of TSS switching event, transcripts use different TSS within the same promoter. In most cases, the TSS switch happened within 100 bp of each other. The TSS switching event does not affect protein coding regions of the gene, but they alter the length of 5′ UTRs. 5′ UTR plays a major role in translational efficiency as 5′ UTR is critical for ribosome recruitment to mRNA and choice of start codon (Hinnebusch et al, 2016). The 5′ UTR contains key elements of translational regulation including structural motifs and uORFs (Jia et al, 2020). These sequence elements in the 5′ UTR contribute to mRNA translation by controlling the selection of translational initiation sites (TIS) (Hinnebusch et al, 2016; Jia et al, 2020). Sequence variation even in only 10 bp immediate upstream of translational start sites or uORFs lead to profound changes in translational efficiency and the amount of protein produced (Dvir et al, 2013; Jia et al, 2020). Thus, a switch of TSS usage between two

strains even within 100 bp could have a profound impact on the protein translation from mRNA. In this study, we identified 425 transcripts with TSS switching between SHR and BN. Furthermore, we show that the genomic variants located in these regions show a significant allelic imbalance in F1s derived by the reciprocal cross of SHR and BN, confirming the TSS switch. TSS switching between SHR and BN might lead to complex disease phenotypes shown by SHR; however, further experimental studies are required to establish a causal role of TSS switching in disease manifestation.

We identified TSS switching in insulin receptor (*Insr*) gene between SHR and BN rat strain resulting in a longer transcript with extended 5′ UTR in SHR compared with BN. The genomic variant located in SHR specific region showed a significant allelic imbalance in F1s, confirming the TSS switch. Furthermore, *Insr* TSS usage was strongly associated with the insulin levels, systolic and diastolic blood pressure in the HXB/BXH panel of recombinant inbred (RI) strains derived by crossing SHR and BN rat strains. The SHR rat strain shows various phenotypes such as hypertension, insulin resistance, dyslipidemia, and central obesity, collectively known as metabolic syndrome (Atanur et al, 2010). The components of metabolic syndrome, hypertension, and insulin resistance often coexist (Zhou et al, 2012, 2014). Clinical studies have shown that 50% of hypertensive individuals have hyperinsulinemia, whereas 80% of type 2 diabetes patients have hypertension (Zhou et al, 2014). It has been shown that the mutations in the insulin receptor gene (*INSR*) lead to severe insulin resistance in humans (Musso et al, 2004; Ros et al, 2015). Down-regulation of insulin receptor is a well-established contributor to insulin resistance (Nagarajan et al, 2016). Previous studies suggest the causal relationship between compensatory hyperinsulinemia due to insulin resistance and hypertension (Zhou et al, 2014; Soleimani, 2015). It is challenging to establish a causal relationship between *Insr* TSS switch and disease phenotypes shown by SHR; however, multiple lines of genetic evidence along with the strong association between *Insr* TSS usage with insulin levels and blood pressure in RI strains suggest that alternate TSS usage might have a phenotypic impact in SHR rat strain.

Taken together, our study suggests that alternative promoter usage may play a role in complex disease phenotypes and paves the way for further experimental studies to establish a causal relationship between alternative promoter usage and complex diseases. Furthermore, data generated in this study could help in improving rat gene annotations, which would significantly benefit researchers who use the rat as a model organism to study complex diseases.

# Materials and Methods

### Rat strains

The rat model of hypertension, SHR/OlaIpcv, and the BN-Lx/Cub rat strains (referred to as a SHR and BN) were maintained in an animal facility at the Institute of Physiology, Czech Academy of Sciences, Prague, Czech Republic. All the experimental procedures were carried out as per European Union National Guidelines and the animal protection laws of the Czech Republic and were approved by the ethics committee of the Institute of Physiology, Czech Academy of Sciences, Prague.

LVs from 6-wk-old SHR (n = 6, three males and three females) and BN (n = 6, three males and three females) were harvested at the Institute of Physiology, Czech Academy of Sciences, Prague. Reciprocal F1 crosses were generated by crossing SHR females with BN males (SHRxBN) and BN females with SHR males (BNxSHR). LVs from si6-wk-old F1s were harvested. We used a total of 12 F1s with six (three males and three females) F1s from each reciprocal cross.

### CAGE tag sequencing

The LVs harvested from rat strains were sent to DNAform, Japan. The RNA extraction and non-tagging, non-amplification (nAnT)-CAGE tag sequencing was performed at DNAform, Japan following a previously described protocol (Murata et al, 2014).

### Read mapping

The CAGE tags were sequenced using 100-bp paired-end sequencing technology on the Illumina HiSeq2500 platform. The adaptors were removed using cutadapt (Martin, 2011). The additional G nucleotide, which is often attached to the 5′ end of the tag by the template free activity of the reverse transcriptase in the cDNA preparation step of the CAGE protocol, was removed. The CAGE tag sequences were mapped to the reference genome using STAR-2.4.0 (Dobin et al, 2013). To avoid read mapping bias due to genomic variants between the SHR and BN rat strains, a pseudo-SHR genome was generated by substituting all SHR single nucleotide variants (Atanur et al, 2010, 2013) from the BN reference genome. The short insertions and deletions (indels) were not substituted to preserve the co-ordinate system between the BN reference genome and the pseudo-SHR genome. CAGE tags from SHR LVs were mapped to the pseudo-SHR reference genome, whereas BN LV CAGE tags were mapped to the BN reference genome (Gibbs et al, 2004). CAGE tags from F1s were mapped to both BN and pseudo-SHR reference genomes and the best hits were selected for the downstream analysis.

### Identification of CAGE-defined tag clusters

All the replicates from parental strains (SHR and BN) were used to identify CAGE-defined tag clusters (TCs). Only uniquely mapped reads, reads mapped with quality score 255 extracted using samtools-1.2 (Li, 2011), were used to identify CAGE-defined tag clusters. All the unique 5′ ends of the first read (R1) of the read pair were considered as a CAGE-defined TSS (CTSS) and counts were generated for each position representing a number of unique tags starting from that position. All the positions where at least three samples showed a minimum of one read count were selected. The CTSSs that were within a 20 bp distance were merged to generate TC. Counts within each tag cluster were then normalised to one million reads (TPM). The TCs that had a normalised read count of 1TPM in at least one sample were selected as a final set. Next, the TCs that were within 100 bp away from each other were clustered together to define the promoter regions.

## Promoter ranking

The gene regions were defined as a genomic region between the start and end position of the longest transcript of a gene plus 1,000 bp upstream of the start position. All the CAGE-defined promoters located within the gene regions on the same strand were assigned to the gene. The promoters were then ranked based on the expression levels (TPM) in descending order. The promoter that showed the highest expression was called P1 and subsequent promoters were called P2, P3, and so on for each gene. All the tag clusters that were located in intergenic regions of the genome were considered independent promoters with a P1 rank.

## Comparison with the human, mouse, and rat CAGE tag sequencing from FANTOM consortium

The CAGE-defined promoter regions from human, mouse, and rat were downloaded from the FANTOM consortium (Forrest et al, 2014; Lizio et al, 2017). The orthologous regions in the human genome (hg19) for the CAGE-defined rat heart promoters identified in this study were identified using the LiftOver tool and rn6tohg19 chain file downloaded from the UCSC genome browser (https://genome.ucsc.edu/). Similarly, orthologous regions in mouse (mm9) were identified using LiftOver tools and corresponding chain files. The human and mouse orthologous regions of rat heart CAGE tag clusters were then overlapped with the human and mouse GACE tag clusters from the FANTOM consortium using bedtools (Quinlan & Hall, 2010).

## Characterisation of CAGE-defined promoters

A list of rat (rn6 assembly) CpG islands was downloaded from the UCSC genome browser (https://hgdownload.soe.ucsc.edu/goldenPath/rn6/database/cpgIslandExtUnmasked.txt.gz). The rat heart CAGE tag clusters extended by 200 bp on both sides were overlapped with the list of CpG islands using bedtools intersectBed (Quinlan & Hall, 2010) to identify CAGE tag clusters that overlap with the CpG islands. To identify TATA-rich CAGE tag clusters, the −500 to 200 bp region around the CAGE peak was scanned to the TATA motif using Homer (Heinz et al, 2010). The Mann–Whitney test was performed in R to determine whether lengths of TATA-rich promoters differ from CpG-rich promoters.

## Identification of genes with alternate promoter usage

To identify genes with alternate promoter usage between SHR and BN, first differentially expressed tag clusters were identified using DEseq2-1.32.0 (Love et al, 2014). The promoters that were differentially expressed (adjusted $P$-value ≤ 0.05) between SHR and BN were extracted. All the promoters that showed at least 20% expression compared with the aggregate expression levels of all the promoters of the same gene were selected for the downstream analysis. The genes where two or more promoters showed differential expression in opposite direction, that is, of the two promoters of the same gene, one promoter showing up-regulation in SHR compared with BN, whereas another showing down-regulation in SHR as compared with BN, were selected. A gene was called to use alternate promoters only when two promoters that showed expression difference in opposite direction included P1 promoter.

## Identification of shifting promoters

To identify switching promoters, for each TC (promoter) the CTSS data (count of unique 5′ ends of CAGE tags within TC at base-pair resolution) from six replicates of each parental strain (SHR and BN) was aggregated. The Kolmogorov–Smirnov test was used to identify switching promoters. Following filtering criteria were used to select statistically significant promoter switching events between SHR and BN. (i) The test statistics $D$ must be greater than 0.3. (ii) The test statistics $D$ must be greater than the critical value at $\alpha$ 0.05. The genomic position where the highest $D$ score was achieved was selected as the position of the switch.

## RNA-seq data analysis

The publicly available RNA-seq data from SHR and BN LV and liver (Johnson et al, 2014) was downloaded from SRA/ENA. Adaptors were removed using cutadapt. RNA-seq reads from the BN were mapped to the BN reference genome, whereas SHR RNA-seq reads were mapped to the pseudo-SHR genome using STAR-2.4.0 (Dobin et al, 2013).

## De novo transcriptome assembly

De novo transcriptome assembly was performed using stringtie-2.1.4 (Pertea et al, 2015) with default parameters using mapped RNA-seq read files (bam files) generated by STAR-2.4.0 (Dobin et al, 2013). De novo assembly was performed independently for SHR and BN transcriptome. In addition, to generate de novo transcripts for rat hearts, transcriptome assemblies from SHR and BN hearts were merged using stringtie-2.1.4 merge.

## Allelic imbalance analysis in F1s

Variants in CAGE-derived tag clusters were identified using GATK-4.0.7 (Depristo et al, 2011; Poplin et al, 2017 *Preprint*) following the recommended guidelines for variant calling from transcriptome data. Reads that spanned intronic regions are mapped to exons by splitting reads. To avoid calling variants in such regions, split reads spanning intronic regions were pre-processed using GATK SplitNCigarReads. Then variants were identified in each sample independently using GATK HaplotypeCaller. The gVCF files were then merged using GATK CombineGVCFs.

Following filtering criteria were used to select variants for allelic imbalance analysis. (i) Variants must be covered with a minimum of 10 reads in all six replicates of both the parental strain. (ii) Variants must be variable between the two parental strains SHR and BN. (iii) In F1, variants must be covered with a minimum of 10 reads in at least nine out of 12 replicates.

To identify the variants that show an allelic imbalance in F1s, reference (BN) allele and alternate (SHR) allele counts were extracted for each high-quality variant. A binomial test was used to identify variants with an allelic imbalance in F1s. The variants that showed $P$-value ≤ 0.05 were called allelically imbalanced variants in F1s.

### RNA-seq data from RI strains

The LV and liver RNA-seq data from RI strains were obtained from the previous publication (Witte et al, 2021).

### *Insr* promoter analysis in RI strains

Based on the genotype at g.1816552:A>G recombinant inbred (RI) strains were grouped into two groups, where AA represents RI strains that inherited the BN allele, whereas GG represents RI strains inherited the SHR allele. RNA-seq read count was obtained from the SHR specific *Insr* transcript region (chr12:1,816,550-1,816,650). The RNA-seq read count from the SHR specific *Insr* transcript region was then normalised to the read counts from the first exon of the *Insr* gene (chr12:1,815,967-1,816,414). The Mann–Whitney test was used to identify a significant association between genotype and normalised read count in SHR specific *Insr* promoter region.

### Phenotypic measurements in RI strains

The phenotypic measurements such as insulin levels, systolic, and diastolic blood pressure in RI strains were obtained from the previous publications (Pravenec et al, 2002; Kuneš et al, 2008). The Mann–Whitney test was used to identify a significant association between genotype and phenotype in RI strains.

### Gene enrichment analysis

Gene enrichment analysis was performed using the Enrichr Web browser (https://maayanlab.cloud/Enrichr/).

## Data Availability

The CAGE data generated in this study have been submitted to the European Nucleotide Archive (ENA; https://www.ebi.ac.uk/ena/browser/home) under accession number PRJEB47228. The codes used for the data analysis are available on GitHub (https://github.com/santoshatanur/ratCAGE). The CAGE peaks identified in this study along with the raw and normalised counts are also available on GitHub (https://github.com/santoshatanur/ratCAGE/tree/main/data).

## Supplementary Information

## Acknowledgements

We acknowledge Professor Piero Carninci, RIKEN, Japan, for the help with the CAGE tag sequencing. We acknowledge Dr Claire Morgan and Dr Goutham Atla for carefully reading the manuscript and providing valuable inputs to improve the manuscript.

## Author Contributions

S Dahale: software and formal analysis.
J Ruiz-Orera: validation (promoter shift in *Insr* gene).
J Silhavy: resources.
N Hübner: validation (promoter shift in *Insr* gene).
S van Heesch: validation (promoter shift in *Insr* gene).
M Pravenec: resources and writing—review and editing.
SS Atanur: conceptualization, software, formal analysis, supervision, funding acquisition, investigation, visualization, methodology, project administration, and writing—original draft, review, and editing.

## Conflict of Interest Statement

The authors declare that they have no conflict of interest.

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
