## [Reviewer comments · Life Science Alliance]

Life Science Alliance

Cap analysis of gene expression reveals alternative promoter usage in a rat model of hypertension.

Sonal Dahale, Jorge Ruiz-Orera, Jan Silhavy, Norbert Hubner, Sebastiaan van Heesch, Michal Pravenec, and Santosh Atanur
DOI: <https://doi.org/10.26508/lsa.202101234>

Corresponding author(s): Santosh Atanur, Imperial College London

Review Timeline:

Submission Date:	2021-09-15
Editorial Decision:	2021-10-26
Revision Received:	2021-12-02
Editorial Decision:	2021-12-22
Revision Received:	2021-12-26
Accepted:	2021-12-28

Scientific Editor: Novella Guidi

Transaction Report:

October 26, 2021

Re: Life Science Alliance manuscript #LSA-2021-01234-T

Santosh S Atanur
Imperial College London, London, UK

Dear Dr. Atanur,

Thank you for submitting your manuscript entitled "Cap analysis of gene expression (CAGE) sequencing reveals alternative promoter usage in complex disease." to Life Science Alliance. The manuscript was assessed by expert reviewers, whose comments are appended to this letter. As you will note from the reviewers' comments below, all the reviewers are quite enthusiastic about the work that in their view is useful to facilitate the analysis of genetic factors underlying the rat hypertension model. However, they do raise some important concerns that need to be addressed before resubmission. Reviewer 1 is concerned that the title is too broad and implies a more global picture than what is reported in the manuscript. It should be focused to a more specific context. Also, it would strengthen the paper if authors would compare the findings in the rat model to promoters in human. The data deposited in ENA aren't accessible, and it would be useful to add also the analysis scripts (for example in github or gitlab) together with CAGE tag cluster coordinates and an expression table (raw counts and normalized expression) to the data deposited. Reviewer 2 requests further characterization of the biological function of the promoter shift in the *Insr* gene. Reviewer 3 main concerns are the lack of proper citation and comparison with previous CAGE-based study on the rat transcriptome by Lizio et al. 2017. Results presented in this paper should be compared to results presented in the previous paper and to corresponding results posted on the FANTOM5 website. We, thus, encourage you to submit a revised version of the manuscript back to LSA that responds to all of the reviewers' points.

Thank you for this interesting contribution to Life Science Alliance. We are looking forward to receiving your revised manuscript.

Sincerely,

-- Summary blurb (enter in submission system): A short text summarizing in a single sentence the study (max. 200 characters including spaces). This text is used in conjunction with the titles of papers, hence should be informative and complementary to the title and running title. It should describe the context and significance of the findings for a general readership; it should be

written in the present tense and refer to the work in the third person. Author names should not be mentioned.

B. MANUSCRIPT ORGANIZATION AND FORMATTING:

Reviewer #1 (Comments to the Authors (Required)):

The manuscript presented by Dahale et al. employs CAGE data from samples of the left heart ventricles of spontaneously hypertensive rats (SHR) and from a normotensive strain (BN). The authors identify gene promoters, how gene expression driven from these promoters differs between SHR and BN with a focus on alternative promoters for genes. 1,970 potentially novel promoter regions in rat heart tissue were identified together with 27 genes displaying alternative promoter usage and 475 promoter switching events between SHR and BN.

The manuscript is well written and the presented results are technically sound. It has to be clearly written, however, that the findings are made for rat heart ventricle samples in SHR and BN rats.

Major issues

The title "... reveals alternative promoter usage in complex disease" implies a more global picture than what is reported in the manuscript. I suggest to rephrase the title to a more specific context "... reveals alternative promoter usage in a rat model of hypertension" or similar.

Do the authors see any opportunity to bioinformatically compare the findings in the rat model to promoters in human?

I was not able to access the dataset from ENA, no dataset with the accession number PRJEB47228 was found.

I suggest to not only provide FASTQ or bam files but also the analysis scripts (for example in github or gitlab) together with CAGE tag cluster coordinates and an expression table (raw counts and normalized expression).

Minor issues

In Figure 4A, it might be helpful to display the SNP position.

Page 3: "Cap analysis of Gene Regulation (CAGE)" should read "Cap Analysis of Gene Expression" or "Cap Analysis Gene Expression".

There are several minor syntax and grammar issues that I recommend to fix.

At least two of the references refer to the same publication, references 5 and 37.

Reviewer #2 (Comments to the Authors (Required)):

In this manuscript, the authors describe the CAGE analysis of the rat model of hypertension. They utilized the spontaneously hypertensive rats (SHR) and analyzed the transcriptional start sites (TSSs) of the left ventricle. For the normal controls, Brown Norway (BN) rats were analyzed. They also analyzed the reciprocal cross F1 offspring rats of SHR and BN to analyze the allele-biased TSS usages. They identified a total of 26,560 TSS, including those of 1,970 novel transcripts. They also found 27 genes showed a clear switch of alternative promoters between SHR and BN rats. Additionally, they identified 475 promoter-"shift" events, in which the TSS clusters were separated by less than 100 bp. Even though the difference was small, the 5'UTRs were changes, which the authors claim have biological consequences. They further compared the TSS patterns of SHR, BN and F1 and found that some of these promoter shift events are allele specific. In particularly, they identified the insulin receptor (Insr)

gene as one of those cases. They confirmed that the blood pressure and hyperinsulinemia due to insulin resistance is associated with the *Insr* promoter shift. Overall, I think this manuscript is well-written. And the obtained data should be useful to facilitate the analysis of genetic factors underlying the rat hypertension model. However, I have to point out that biological characterization of the generated catalogue of TSSs should be shallow.

Major points:

1. Further characterization should be needed on the biological function of the promoter shift in the *Insr* gene. It is discussed that the elements of 5'UTR may change, but it remains elusive whether the changed part contain some functional elements.
2. I wonder whether there is any counterpart phenomenon in the *Insr* gene in humans?

Minor points:

3. Reproducibility of the data should be described more clearly. I assume they constructed multiple CAGE libraries per sample, even though their precise information is not given in the text.
4. Please note the tissues they analyzed are "bulk". These days, single cell transcriptome data is widely available, at least in mice. The discussion may include in what cell types in the left ventricle the observed TSSs may take place.
5. I did not understand why the change in the receptor gene of insulin would lead to the observed symptoms. Is the level of the receptor the rate limiting step to the insulin signal?
6. I assume there is a substantial amount of CAGE data in humans and mice, also in hearts. What proportion of the alternative promoters and their changes are conserved with rats?
7. For the promoter shift events, some examples should be presented to show the shapes are merged with SHR and BN in their F1. I wonder whether allele-specific events are mutually independent. It is also important to show they are not the library construction bias.
8. The statistics shown in Supplementary Tables 1 and 2 seems discrepant to the numbers shown in the text.
9. Please also check, Figure 2A describes 28 genes, although the text describes 27 genes.
10. Scale of TPM axis seems arbitrary (not always starting from 0) in Figures in general.

Reviewer #3 (Comments to the Authors (Required)):

This paper presents interesting results on alternative promoter usage in a rat model of hypertension. The data were obtained with the powerful CAGE technology, capable of producing genome-wide TSS maps (promoteromes) at single base resolution. Equally important in my opinion is the re-analysis of public RNA-seq data by the authors, which led to the discovery of thousands of novel transcripts in two rat strains (novel with respect to ENSEMBL annotation). This highlights the need for more systematic efforts to experimentally characterize and computationally infer the tissue-specific transcript diversity in model organisms other than human and mouse.

The results described in this paper are deposited in ENA, and are definitely highly valuable. The paper is well written. The work is technically sound.

Some of the claims may be debatable. Most importantly, the authors are not the first to apply CAGE to rat tissue (see below). The following novelty statement from this paper is thus false: "to date the precise TSS for rat genes have not been mapped."

The fact that alternative promoter usage is observed in differentially expressed genes doesn't necessarily justify a claim of the following kind: "Approximately, 80% of the CAGE defined TSS showed tissue-specific expression, suggesting that the majority of mammalian genes use alternative promoters in a tissue-specific manner to regulate tissue-specific gene expression". There is broad consensus that much of the observed intra- and interspecies transcriptional variation is non-functional. Moreover, the above cited sentence implies design. It is debatable whether something that goes wrong in disease (or something that happens in a rat strain susceptible to disease) can tell us something about design principles of a healthy organisms.

Furthermore, the following statement "this study could be the first step towards the understanding role (sic!) of alternative promoter usage in complex disorders such as hypertension" should perhaps also be toned down in view of the substantial corpus of literature on alternative promoter usage in cancer (see for instance Deniz Demircioğlu et al. 2019, 10.1016/j.cell.2019.08.018).

Major:

The previous CAGE-based study on the rat transcriptome by Lizio et al. 2017 (<https://doi.org/10.1038/sdata.2017.173>) needs to be cited. Wherever possible, results presented in this paper should be compared to results presented in the previous paper and to corresponding results posted on the FANTOM5 website (https://fantom.gsc.riken.jp/5/datafiles/latest/extra/CAGE_peaks/). In particular, the TSS peak list obtained by the authors should be compared to the rat FANTOM5 TSS peak list, and reported novelty counts should be down-scaled accordingly. I noted for instance one of the novel heart-specific Mef2c TSS peaks (Figure 1C) claimed to be novel, overlaps with three shorter FANTOM5 TSS peaks, see coordinates below:

This paper Table S1 (promoter Mef2c_P1):

chr2..11678617..11678804+

FANTOM5 (3 TSS peaks):

chr2:11678649..11678650+,

chr2:11678652..11678663+,

chr2:11678678..11678704+

Note that the FANTOM5 CAGE data can be visualized at the UCSC genome browser by connecting to the public FANTOM5 track hub at:

<https://genome-euro.ucsc.edu/cgi-bin/hgHubConnect>

Minor:

Method section: The mapping of the reads to two alternative genome is not fully explained. I surmise that the pseudo-SHR genome exhibits some indel variants relative to the wild-type genome. This would have the consequence, that homologous genome positions have different coordinates in the two genomes. Somehow, TSS coordinates from the SHR genome would have to be mapped back to the BN genome for comparative promoter usage analysis. This should be explained in enough detail such that interested readers could reproduce the analysis. Alternatively, if the computational recipe has already been published elsewhere, a citation would do so.

Figure 3A: Indicate the exact genomic coordinates (to enable the reader to look at the corresponding FANTOM5 CAGE data in the UCSC genome browser)

Discretionary:

The Mef3c_P1 promoter mentioned above coincides with the 5'end of a homologous transcript in mouse (observation made via the UCSC genome browser). It would be interesting to know how many of the 7934 "novel" TSS (Fig. 1B) coincide with the 5'ends of homologous mouse transcripts.

Regarding the genomic variant "g.186552:A>G", which is located in the SHR-specific extension of the Insr transcript: It not clear why the search for potentially causal variants was restricted to this very short genomic region. Variants influencing the choice of a TSS may be located at larger distances upstream or downstream of the selected TSS. Are there other interesting variants in the neighborhood of g.186552:A>G, which could be responsible for the TSS shift?

Responses to reviewers

We thank all the reviewers for the constructive comments to improve the overall quality of the manuscript. We went through all the reviewers' comments carefully and addressed them to the best of our ability. We performed two new analyses to address concerns of the reviewers. i) Systematic comparison of rat heart CAGE tag clusters with the human and mouse (Forrest et al. 2014) as well as the rat CAGE data from other cell types (Lizio et al. 2017) generated by FANTOM consortium. ii) Systematic characterisation of rat heart CAGE defined promoters. In addition, we have provided all the scripts used for analysis of the rat heart CAGE data on GitHub along with the raw and normalised counts for the tag clusters.

Below we provide our point-to-point responses and hope that they are satisfactory.

Reviewer 1:

Major issues

Comment 1.1: The title "... reveals alternative promoter usage in complex disease" implies a more global picture than what is reported in the manuscript. I suggest to rephrase the title to a more specific context "... reveals alternative promoter usage in a rat model of hypertension" or similar.

Response 1.1: As suggested by the reviewer, the title of the manuscript is changed to "Cap analysis of gene expression (CAGE) sequencing reveals alternative promoter usage in a rat model of hypertension" to make it more specific.

Comment 1.2: Do the authors see any opportunity to bioinformatically compare the findings in the rat model to promoters in human?

Response 1.2: We have performed a systematic comparison between rat heart CAGE data generated in this study with the human and mouse cage data from the FANTOM consortium. We have added a new section in the revised manuscript on the comparison of rat CAGE data with human and mouse CAGE data. (Page 6, lines 2 to page 7, lines 2). Corresponding methods are included in the Methods section.

Comment 1.3: I was not able to access the dataset from ENA, no dataset with the accession number PRJEB47228 was found.

Response 1.3: The data has been submitted to ENA with accession number PRJEB47228. It was under embargo until publication, which is a common practice, hence it was not accessible publicly. Data was expected to be released as soon as the manuscript gets accepted for a publication. However, on reviewer's request we have removed an embargo and data will be made public on 3rd December 2021. Hopefully, now reviewer can access the data using accession number PRJEB47228.

Here are some screenshots of ENA to prove that data was indeed submitted to ENA under accession PRJEB47228.

- 1) Project entry

Primary Accession	Secondary Accession	Title	Submission Date	Status	Release Date	Edit
PRJEB47228	ERP131490	Cap analysis of gene expression (CAGE) tag sequencing from the left ventricle (LV) of a rat model of [...]	27-Aug-2021	Confidential	03-Dec-2021	Edit

2) Sample entries

Primary Accession	Secondary Accession	Title	Tax ID	Scientific Name	Common Name	Submission Date	Status	Edit
ERS7278272	SAMEA955521	BN female CAGE R1	10116	Rattus norvegicus	Norway rat	27-Aug-2021	Confidential	Edit
ERS7278271	SAMEA955520	BN male CAGE R3	10116	Rattus norvegicus	Norway rat	27-Aug-2021	Confidential	Edit
ERS7278270	SAMEA955519	BN male CAGE R2	10116	Rattus norvegicus	Norway rat	27-Aug-2021	Confidential	Edit
ERS7278269	SAMEA955518	BN male CAGE R1	10116	Rattus norvegicus	Norway rat	27-Aug-2021	Confidential	Edit
ERS7278268	SAMEA955517	SHR female CAGE R3	10116	Rattus norvegicus	Norway rat	27-Aug-2021	Confidential	Edit
ERS7278267	SAMEA955516	SHR female CAGE R2	10116	Rattus norvegicus	Norway rat	27-Aug-2021	Confidential	Edit
ERS7278266	SAMEA955515	SHR female CAGE R1	10116	Rattus norvegicus	Norway rat	27-Aug-2021	Confidential	Edit
ERS7278265	SAMEA955514	SHR male CAGE R3	10116	Rattus norvegicus	Norway rat	27-Aug-2021	Confidential	Edit
ERS7278264	SAMEA955513	SHR male CAGE R2	10116	Rattus norvegicus	Norway rat	27-Aug-2021	Confidential	Edit
ERS7278263	SAMEA955512	SHR male CAGE R1	10116	Rattus norvegicus	Norway rat	27-Aug-2021	Confidential	Edit
ERS7278274	SAMEA955523	BN female CAGE R3	10116	Rattus norvegicus	Norway rat	27-Aug-2021	Confidential	Edit
ERS7278273	SAMEA955522	BN female CAGE R2	10116	Rattus norvegicus	Norway rat	27-Aug-2021	Confidential	Edit

Comment 1.4: I suggest to not only provide FASTQ or bam files but also the analysis scripts (for example in github or gitlab) together with CAGE tag cluster coordinates and an expression table (raw counts and normalized expression).

Response 1.4: As suggested by reviewer, the Perl codes used for the analysis of the CAGE data are provided on GitHub (<https://github.com/santoshatanur/ratCAGE>). Along with the codes, we have also provided demo data so that codes can be tested. Furthermore, as suggested by reviewer, we have provided a list of CAGE tag clusters identified in this study along with the raw counts and normalised counts (TPM) on GitHub under the directory named "data".

Minor issues

Comment 1.5: In Figure 4A, it might be helpful to display the SNP position.

Response 1.5: The SNP position is now added to Figure 4A.

Comment 1.6: Page 3: "Cap analysis of Gene Regulation (CAGE)" should read "Cap Analysis of Gene Expression" or "Cap Analysis Gene Expression".

Response 1.6: Thank you very much for pointing out this mistake, we have now changed it. (Page 3, line15)

Comment 1.7: There are several minor syntax and grammar issues that I recommend to fix.

Response 1.7: We have tried to correct all the syntax errors and grammar mistakes to the best of our ability.

Comment 1.8: At least two of the references refer to the same publication, references 5 and 37.

Response 1.8: Thank you for bringing this to our notice. We have corrected this and a few other reference duplications.

Reviewer 2:

Major points:

Comment 2.1: Further characterization should be needed on the biological function of the promoter shift in the *Insr* gene. It is discussed that the elements of 5'UTR may change, but it remains elusive whether the changed part contain some functional elements.

Response 2.1: We have performed systematic characterisation of not only *the Insr* promoter but all the rat heart CAGE tag clusters. We have included a new sub section in the revised manuscript on CAGE tag cluster characterisation under the title “**Characterisation of rat heart CAGE tag clusters**” (Page 7, lines 4-22).

We also characterised *Insr* promoter in detail. We show that the *Insr* promoter show the signature of a housekeeping gene as *Insr* promoter is CpG-rich promoter (Page 14, lines 5-8).

Comment 2.2: I wonder whether there is any counterpart phenomenon in the *Insr* gene in humans?

Response 2.2: We are not aware of any study that specifically accessed the promoter shift in individuals with metabolic syndrome or hypertension. However, FANTOM CAGE data suggests that both the SHR and BN promoters are conserved in humans.

Minor points:

Comment 2.3: Reproducibility of the data should be described more clearly. I assume they constructed multiple CAGE libraries per sample, even though their precise information is not given in the text.

Response 2.3: We have sequenced CAGE tags from three biological replicates of each sample. We estimated the reproducibility and the results are reported in the revised manuscript (Page 5, line 14-18).

“To assess the reproducibility of the experiments we estimated all possible pairwise correlations between the three biological replicates of each sample (SHR male, SHR female, BN male, and BN female). The biological replicates of each sample showed a pairwise Pearson correlation coefficient of 0.98 or higher (Supplemental Figure 1), suggesting that our results are highly reproducible.”

Additionally, we have included a supplemental figure to show reproducibility between the replicates.

Comment 2.4: Please note the tissues they analyzed are "bulk". These days, single cell transcriptome data is widely available, at least in mice. The discussion may include in what cell types in the left ventricle the observed TSSs may take place.

Response 2.4: In absence of single cell data, it is challenging to speculate cell type specificity of the CAGE tag signals. However, we have included the following paragraph in the discussion. (Page 16, lines 22 to page 17, lines 3).

“Rat heart CAGE data has been generated from the whole left ventricle. The heart is a complex organ with various cell types (Talman & Kivelä, 2018). In absence of single-cell transcriptomic data from rat hearts, it is challenging to determine cell type specificity of the CAGE tag signal. However, as cardiomyocytes and endothelial cells are the two most abundant cardiac cell types (Talman & Kivelä, 2018) and human heart single-cell transcriptomic data suggest the 50% of the ventricular cells tend to be cardiomyocytes (Litviňuková et al., 2020), we speculate that majority of the CAGE defined TSSs might be active in cardiomyocytes. However, small proportions of TSS signals might also be coming from other rat left ventricular cell types such as endothelial cells, fibroblast, pericytes, and smooth muscle cells.”

Comment 2.5: I did not understand why the change in the receptor gene of insulin would lead to the observed symptoms. Is the level of the receptor the rate limiting step to the insulin signal?

Response 2.5: Multiple studies suggest that the optimal level of the insulin receptor is critical for the proper functioning of an insulin signaling pathway. To make it clearer we have included the following lines in a discussion. (Page 18, lines 20-22).

“It has been shown that the mutations in the insulin receptor gene (*INSR*) lead to severe insulin resistance in humans (Musso et al. 2004; Ros et al. 2015). Downregulation of insulin receptor is a well-established contributor to insulin resistance (Nagarajan et al. 2016).”

Comment 2.6: I assume there is a substantial amount of CAGE data in humans and mice, also in hearts. What proportion of the alternative promoters and their changes are conserved with rats?

Response 2.6: We have performed a systematic comparison between rat heart CAGE data generated in this study with the human and mouse cage data from the FANTOM consortium. We have added a new section in the revised manuscript on the comparison of rat CAGE data with human and mouse CAGE data (Page 6, lines 2 to page 7, lines 2). Corresponding methods are included in the Methods section.

Additionally, we also estimated the conservation of promoter regions of the genes that show alternative promoter usage between SHR and BN with humans and mouse. Results are included in the revised manuscript in corresponding sections.

Alternative promoter usage (Page 11, lines 11-15): “Of 28 genes that use alternative promoter between SHR and BN in the heart, both the alternative promoters of 10 genes were conserved in humans while only one of the two promoters of eight genes were conserved in humans. Similarly, both the alternative promoters of 13 genes were conserved in mouse while for nine genes only one promoter was conserved in mouse.”

Shifting promoters (Page 12, lines 16-22): “Of 475 heart transcripts that showed a shift in the TSS usage between SHR and BN, human orthologous regions of 182 promoters showed CAGE tag signals in FANTOM data. Of 182 transcripts, for 89 transcripts only one of the two (SHR or BN) promoters was represented in humans, while both SHR and BN promoters were represented in humans for 93 transcripts. Similarly, a total of 200 rat shifting promoters were conserved in mouse. In mouse, 144 transcripts showed both SHR and BN promoters while 56 mouse transcripts showed only of the two (SHR or BN) promoters.”

Comment 2.7: For the promoter shift events, some examples should be presented to show the shapes are merged with SHR and BN in their F1. I wonder whether allele-specific events are mutually independent. It is also important to show they are not the library construction bias.

Response 2.7: We have included a supplemental figure to show merged shape in F1 for three genes (*Insr*, *Trna1ap*, and *Vnn1*) that show promoter shift between SHR and BN. In F1s both the SHR and BN promoters show activity, and the expression levels were average of BN and SHR expression levels at both the promoters. This suggests that the promoter shift events and the allelic expression are not mutually independent. We have included the following text in the revised manuscript.

“The F1s inherit both longer and shorter transcripts of the genes that show promoter switching between the two rat strains thus both the promoters show intermediate expression in F1s (Supplemental Figure 4).” Page 12, line 28 to page 13, line 2.

Comment 2.8: The statistics shown in Supplementary Tables 1 and 2 seems discrepant to the numbers shown in the text.

Response 2.8: A total of 857 genic tag clusters (with no ENSEMBL TSS annotations) had support from at least one *de novo* transcript TSS. However, a total of 12 tag clusters had support from two independent *de novo* transcripts TSSs. In Supplemental Table1, 12 rat CAGE tag clusters have duplicate entry to accommodate overlap with two independent *de novo* transcript TSS, thus in Supplemental Table1, there are 869 lines, which represent 857 rat CAGE tag clusters.

Similarly, a total of 1,113 intergenic rat CAGE tag clusters have *de novo* transcript TSS support, with 13 rat CAGE tag clusters overlapping two independent *de novo* transcript TSSs, hence we have 1,126 entries in Supplemental Table2.

Comment 2.9: Please also check, Figure 2A describes 28 genes, although the text describes 27 genes.

Response 2.9: Thank you very much for pointing this. We have changed the numbers throughout the manuscript to match with the numbers shown in the figure.

Comment 2.10: Scale of TPM axis seems arbitrary (not always starting from 0) in Figures in general.

Response 2.10: We used the default setting (autoscaling) of the UCSC browser while generating these plots for the original manuscript. However, in the revised manuscript we have changed it so that the TPM axis always starts from 0, consistently across all the figures.

Reviewer 3:

General Comments:

Comment 3.1: Some of the claims may be debatable. Most importantly, the authors are not the first to apply CAGE to rat tissue (see below). The following novelty statement from this paper is thus false: "to date the precise TSS for rat genes have not been mapped."

Response 3.1: We thank the reviewer for bringing it to our notice that CAGE tag sequencing has been performed in rats albeit in different cell types and strains than the one used in this study. We have rephrased the above sentence in the revised version of the manuscript to "TSS for rat genes have been mapped only in three cell types (Aortic smooth muscle cells, hepatocytes, and mesenchymal stem cells) and a tissue". Page 16, lines 8-10.

Comment 3.2: Furthermore, the following statement "this study could be the first step towards the understanding role (sic!) of alternative promoter usage in complex disorders such as hypertension" should perhaps also be toned down in view of the substantial corpus of literature on alternative promoter usage in cancer (see for instance Deniz Demircioğlu et al. 2019, 10.1016/j.cell.2019.08.018).

Response 3.2: Thank you very much for pointing towards the literature on alternative promoter usage in cancer. In the revised manuscript, we have removed the above sentence.

However, we would like to clarify that, our intention was not to claim that this is the first study that showed alternative promoter usage in disease. We intended to highlight the fact that "results presented in this manuscript are based on computational analysis. Computational analysis is the first step in the identification of alternative promoter usage and extensive experimental analysis is required to confirm these findings".

Major comments:

Comment 3.3: The previous CAGE-based study on the rat transcriptome by Lizio et al. 2017 (<https://doi.org/10.1038/sdata.2017.173>) needs to be cited. Wherever possible, results presented in this paper should be compared to results presented in the previous paper and to corresponding results posted on the FANTOM5 website (https://fantom.gsc.riken.jp/5/datafiles/latest/extra/CAGE_peaks/). In particular, the TSS peak list obtained by the authors should be compared to the rat FANTOM5 TSS peak list, and reported novelty counts should be down-scaled accordingly. I noted for instance one of the novel heart-specific Mef2c TSS peaks (Figure 1C) claimed to be novel, overlaps with three shorter FANTOM5 TSS peaks, see coordinates below:

This paper Table S1 (promoter Mef2c_P1):

chr2..11678617..11678804+
FANTOM5 (3 TSS peaks):
chr2:11678649..11678650+,
chr2:11678652..11678663+,
chr2:11678678..11678704+

Note that the FANTOM5 CAGE data can be visualized at the UCSC genome browser by connecting to the public FANTOM5 track hub at:

<https://genome-euro.ucsc.edu/cgi-bin/hgHubConnect>

Response 3.3: We are extremely sorry for not citing the previous paper on rat CAGE by Lizio et al 2017. We are thankful to the reviewer for bringing this to our notice. In the revised manuscript, we have cited this paper wherever appropriate.

We performed a systematic comparison between the rat heart CAGE data from this study and the FANTOM consortium rat CAGE data (Lizio et. al. 2017) along with human and mouse CAGE data. We have included a new sub-section titled “**Comparison with the human, mouse and rat CAGE data from FANTOM consortium**” in results (Page 6, line 2 to page 7, line 2) and a corresponding section in Methods (Page 21, line 20 to page 22, line 3).

Additionally, we changed following subheading in result section “**CAGE tag sequencing identifies novel cardiac TSS/transcripts**” to “**CAGE tag sequencing improves rat TSS/transcripts annotations**” in revised manuscript to avoid false novelty claims. Please note that all our claims were in comparison to ENSEMBL annotations.

Furthermore, we compared genic (overlapping with gene region) rat heart CAGE tag clusters with no ENSEMBL TSS annotations with the FANTOM rat CAGE data, and results are presented in the appropriate sub-section (Page 8, lines 14-16).

“Next, we investigated whether rat heart CAGE tag clusters without ENSEMBL TSS annotations were represented in other rat cell types/tissues, human or mouse. Of 7,934 tag clusters, a small proportion of tag clusters (n=730) were also present in FANTOM rat CAGE data.”

Additionally, we compared intergenic rat heart CAGE tag clusters with the FANTOM rat CAGE data, and results are presented on page 9, line 25 to 28.

“Furthermore, 3,054 (63.66%) rat heart intergenic CAGE tag clusters overlapped with the CAGE defined promoters either from other rat tissues (n=1,229), mouse heart promoters (n=2,578), or human heart promoters (n=1,832) when compared to FANTOM CAGE dataset.”

Minor:

Comment 3.4: Method section: The mapping of the reads to two alternative genome is not fully explained. I surmise that the pseudo-SHR genome exhibits some indel variants relative to the wild-type genome. This would have the consequence, that homologous genome positions have different coordinates in the two genomes. Somehow, TSS coordinates from the SHR genome would have to be mapped back to the BN genome for comparative promoter usage analysis. This should be explained in enough detail such that interested readers could reproduce the analysis. Alternatively, if the computational recipe has already been published elsewhere, a citation would do so.

Response 3.4: We would like to clarify that, we substituted only single nucleotide variants (SNVs) to keep the co-ordinate system intact between BN and pseudo-SHR genome. We have not substituted indels. To make this point clear we have updated the methods section. To make it clearer we have revised the methods.

“To avoid read mapping bias due to genomic variants between the SHR and BN rat strains, a pseudo-SHR genome was generated by substituting all SHR single nucleotide variants (SNVs) (Atanur et al., 2010, 2013) from the BN reference genome. The short insertions and deletions (indels) were not

substituted to preserve the co-ordinate system between the BN reference genome and the pseudo-SHR genome.”

Comment 3.5: Figure 3A: Indicate the exact genomic coordinates (to enable the reader to look at the corresponding FANTOM5 CAGE data in the UCSC genome browser)

Response 3.5: We have updated Figure 3A to include exact genomic coordinates.

Discretionary:

Comment 3.6: The Mef3c_P1 promoter mentioned above coincides with the 5'end of a homologous transcript in mouse (observation made via the UCSC genome browser). It would be interesting to know how many of the 7934 "novel" TSS (Fig. 1B) coincide with the 5'ends of homologous mouse transcripts.

Response 3.6: As mentioned above, we performed a systematic comparison between the rat heart CAGE data from this study and the FANTOM consortium rat CAGE data (Lizio et. al. 2017) along with human and mouse CAGE data. We have included a new sub-section titled **“Comparison with the human, mouse and rat CAGE data from FANTOM consortium”** in results (Page 6, line 2 to page 7, line 2) and a corresponding section in Methods (Page 21, line 20 to page 22, line 3).

Additionally, we compared the novel genic as well as intergenic tag clusters with human and mouse CAGE data. In the revised manuscript, novelty claims were down-scaled accordingly. (Page 8, lines 14-16 and page 9, line 25 to 28).

Comment 3.7: Regarding the genomic variant "g.186552:A>G", which is located in the SHR-specific extension of the Insr transcript: It not clear why the search for potentially causal variants was restricted to this very short genomic region. Variants influencing the choice of a TSS may be located at larger distances upstream or downstream of the selected TSS. Are there other interesting variants in the neighborhood of g.186552:A>G, which could be responsible for the TSS shift?

Response 3.7: We completely agree with the reviewer that the causal variants might be anywhere in the genomic region around the Insr gene. The variant 186552:A>G may or may not be a causal variant. However, as the variant was located within the SHR extended region, it was used to get allelic counts in F1 and RI strains to calculate allelic imbalance only.

December 22, 2021

RE: Life Science Alliance Manuscript #LSA-2021-01234-TR

Dr. Santosh S Atanur
Imperial College London
Department of Metabolism, Digestion and Reproduction
5th Floor ICTEM Building
Hammersmith Hospital Campus
London W12 0NN
United Kingdom

Dear Dr. Atanur,

Thank you for submitting your revised manuscript entitled "Cap analysis of gene expression reveals alternative promoter usage in a rat model of hypertension.". We would be happy to publish your paper in Life Science Alliance pending final revisions necessary to meet our formatting guidelines.

- please note that titles in the system and the manuscript file must match
- please add callouts for Figures S1A-D, S3A-B, S4A-C to your main manuscript text

A. FINAL FILES:

B. MANUSCRIPT ORGANIZATION AND FORMATTING:

Sincerely,

Reviewer #1 (Comments to the Authors (Required)):

My comments were adequately addressed.

Reviewer #2 (Comments to the Authors (Required)):

First of all, I appreciate the substantial efforts of the authors for the revision of this manuscript. I believe, with the several lines of extensive analyses and the deepened discussion, this manuscript has been very much improved. I still have a remaining concern about the biological relevance of their findings, especially when considering the data has been obtained from the heterogeneous cellular populations. However, I also consider that this issue may be further pursued as their future studies. In fact, I sincerely hope the authors should continue their efforts to make even better use of this rat model to eventually overcome this difficult disease.

Reviewer #3 (Comments to the Authors (Required)):

The authors have taken the criticisms of all three reviewers very seriously and have added a lot of very valuable new material to the submission. I have no further comments or requests.

December 28, 2021

RE: Life Science Alliance Manuscript #LSA-2021-01234-TRR

Dr. Santosh S Atanur
Imperial College London
Department of Metabolism, Digestion and Reproduction
5th Floor ICTEM Building
Hammersmith Hospital Campus
London W12 0NN
United Kingdom

Dear Dr. Atanur,

Thank you for submitting your Research Article entitled "Cap analysis of gene expression reveals alternative promoter usage in a rat model of hypertension.". It is a pleasure to let you know that your manuscript is now accepted for publication in Life Science Alliance. Congratulations on this interesting work.

DISTRIBUTION OF MATERIALS:

Again, congratulations on a very nice paper. I hope you found the review process to be constructive and are pleased with how the manuscript was handled editorially. We look forward to future exciting submissions from your lab.

Sincerely,
